# The Effect of Combined Atmospheric Plasma/UV Treatments on Improving the Durability of Flame Retardants Applied to Cotton

**DOI:** 10.3390/molecules27248737

**Published:** 2022-12-09

**Authors:** Maram Ayesh, Arthur Richard Horrocks, Baljinder K. Kandola

**Affiliations:** IMRI, School of Engineering, University of Bolton, Bolton BL3 5AB, UK

**Keywords:** atmospheric plasma, UV laser, cotton, flame retardant, diammonium phosphate, urea, 3-aminopropyltriethoxy silane, diethyl N,N bis (2-hydroxyethyl) aminomethylphosphonate, surface treatment, textile, scanning electron microscopy, X-ray photoelectron spectroscopy, thermal analysis

## Abstract

Application of a combined atmospheric plasma/UV laser to cotton fabrics impregnated with selected non-durable flame retardants (FRs) has shown evidence of covalent grafting of the latter species on to cotton fibre surfaces. As a result, an increase in their durability to water-soaking for 30 min at 40 °C has been recorded. Based on previous research plasma gases comprising Ar^80%^/CO_2_^20%^ or N_2_^80%^/O_2_^20%^ were used to pre-expose cotton fabric prior to or after FR impregnation to promote the formation of radical species and increased –COOH groups on surface cellulosic chains, which would encourage formation of FR-cellulose bonds. Analysis by scanning electron microscopy (SEM/EDX), X-ray photoelectron spectroscopy (XPS) and thermal analysis (TGA) suggested that organophosphorus- and nitrogen- containing flame retarding species in the presence of the silicon-containing molecules such as 3-aminopropyltriethoxy silane (APTS) resulted in formation of FR-S-O-cellulose links, which gave rise to post-water-soaking FR retentions > 10%. Similarly, the organophosphorus FR, diethyl N, N bis (2-hydroxyethyl) aminomethylphosphonate (DBAP), after plasma/UV exposure produced similar percentage retention values possibly via (PO).O.cellulose bond formation, While none of the plasmas/UV-treated, FR-impregnated fabrics showed self-extinction behaviour, although burning rates reduced and significant char formation was evident, it has been shown that FR durability may be increased using plasma/UV treatments.

## 1. Introduction

Recent interest in the use of plasma technologies applied to textile fabric surfaces in order to confer a number of novel properties has recently been reviewed [1,2,3]. However, the potential for conferring flame retardancy and especially to cotton has been less researched [2,4,5,6,7,8,9,10,11], although of these the more desirable use of atmospheric plasma either as cold flame plasma jet [8,9] or dielectric barrier discharge (DBD) applied across open width fabric widths [10] has been of more recent focus. One significant challenge in endeavouring to confer flame retardancy is the need for relatively high concentrations of active flame retardant species and their location on activated fibre surfaces [11,12].

Atmospheric plasma exposure has been shown to activate a number of fibre surfaces [1,2,3] and in particular, cotton on which removal of waxes has been observed [9,13,14] as well as modifications of the surface chemistry [15,16,17,18]. Improvements in such properties as hydrophilicity, while dependent on plasma power and gaseous environment for plasma jet [19] and dielectric barrier discharge (DBD) atmospheric exposure times [15,16,18], apart from surface wax removal, have promoted increases in surface functionalities such as carbonyl and carboxylic acid groups under oxidising plasma gas conditions. Our recently published work, which investigated the effects of DBD plasma exposure in the presence and absence of an incident UV 308nm excimer laser [18,20], showed that addition of a low (20%) concentration of carbon dioxide in the nitrogen plasma gas, while producing negligible changes in oxygen-containing species under the combined plasma/laser condition, O/C ratios increased slightly especially when the laser was absent. This latter sample also showed the greatest wettability. Methylene blue dye sorption increased following plasma/UV exposures, especially when the plasma gas was N_2_^80%^/O_2_^20%^, although only “with UV laser” samples were analysed. These results suggested the formation of carboxylic acid groups on cotton fibre surfaces had occurred. While under nitrogen plasma gas environments other workers had reported the formation of N-containing species observed using EDX studies [21], no changes in trace nitrogenous species were observed in our work using XPS. A similar situation with regard to the effect of oxidising plasma atmospheres was observed when polyamide 6.6 fabrics were exposed to the same plasma/UV laser system [22] under a variety of gases (argon, nitrogen and mixtures with oxygen or carbon dioxide) which also created changes to fibre surface chemistry in the form of additional functional groups, principally -NH_2_ and -COOH.

Clearly plasma exposure, especially in oxidising atmospheres besides showing evidence of changes in surface topography as a consequence of etching [18] promotes the formation of potentially reactive surface groups which could react with flame retardant species introduced before, during or after plasma treatments to encourage their grafting to fibre surfaces and thus increase their durability to cleansing treatments. Whereas grafting of additional species following or during plasma exposure on to polymer surfaces is well documented [23,24,25], attempts to confer flame retardancy have been less so. While there has been a number of studies using low pressure plasma, reviewed elsewhere [26], of relevance to this study are those relating to plasma flame or atmospheric plasma only [8,27].

Earlier work in our own laboratories [8] showed that plasma flame exposure of already flame retardant fabrics such as Proban^®^-treated cotton [28] and meta-aramid (Nomex) fabrics impregnated with functionalized clay particles, further enhanced their respective resistance to ignition. It was concluded that strong binding forces may have occurred between the respective fibre surfaces and deposited clay particles after plasma treatment, not only providing a flame shield, but also one that was resistant to a hot water soaking test. Furthermore, because hexamethyldisiloxane (HMDSO) monomer had been shown in plasma-enhanced, chemical vapour deposition (PECVD) [29] to polymerize on the substrate and form a polysiloxane layer on fibre or polymer surfaces, its introduction to the argon gas plasma flame was also studied and shown to further improve flame retardancy [8]. The role of the added poly(HMDSO) was two-fold, namely the enhancement of water repellent properties [30] and the increased surface silicon content on fibre surfaces, which would increase the fire barrier properties.

Specifically with regard to cotton, Edwards et al. [31] have grafted two types of phosphoramidate flame-retardant monomers with different nitrogen percentages on to cotton fabric by post-DBD plasma treatment. While thermal analytical (TGA) responses of grafts under nitrogen showed changes in the thermal behaviour of the fabric in terms of increased char, vertical strip burning tests showed that the samples coated with the higher percentage of nitrogen-containing monomer had decreased the burning rate with no afterglow, while the monomer with lower nitrogen content had increased the burning rate and again showed no afterglow. However, the percentage of phosphorus present after plasma treatment was very low, and the samples were not self-extinguishing. They suggested that the phosphorus-containing fraction of the molecule was degraded by the plasma and washed off during the 12 h Soxhlet extraction.

With such observations in mind, this current work aims to investigate the possibility of increasing the wash durability of a number of phosphorus-, nitrogen- and/or silicon-containing flame retardant (FR) species applied to cotton subjected to pre-plasma/UV laser-exposed fabric or after FR application, post-plasma/UV exposure or a combination or the two sequences. Plasma/UV–exposed fabrics were characterised by scanning electron microscopy (SEM), photoelectron spectroscopy (XPS), thermal analysis and flammability testing using vertical strip and limiting oxygen index techniques before and after a water soaking method. This last comprises an immersion of fabrics in water at 40 °C for 30 min as defined in the current UK furniture regulations [32,33].

## 2. Results

It is noteworthy to observe that subjectively none of the plasma- or plasma/UV-exposed fabrics appeared to have changed from their initially white appearance.

### 2.1. Effect of Plasma on Flame Retardant Retention following 40 °C Water-Soaking

#### 2.1.1. Effect of Plasma as a Post-Treatment on Fabrics Impregnated with FRs Comprising Combinations of DAP, Urea, TEOS, GM and APTS at a Plasma Power of 100 W/m^2^·min

Table 1 shows the dry add-on percent values, which were calculated after the addition of each flame retardant formulation; no change is assumed in each value during plasma treatment, although there is always the possibility of a small reduction due to surface etching. For instance, pure cotton fabric (Cot) shows a reduction in mass after water-soaking of around 4%, hence there was removal of some impurities still present on the commercial bleached fabric, which could include traces of sodium salts following neutralisation after scouring and bleaching. After plasma treatment (sample Cot_PL(N_2_/O_2_) the percent of mass loss after water-soaking has increased to 6%, possibly due to the presence of oxygenated species in the plasma gas, that may have given rise to formation of small water-soluble species such as hemi-celluloses [13]. The amount of FR remaining after water-soaking was calculated using Equation (3) (see Section 4.5).

For all non-plasma-exposed samples after water-soaking there appears to be residual amounts of each flame-retardant combination present in spite of their being considered to be non-durable to aqueous cleansing treatments. Cotton treated with DAP-urea and 3-aminopropyltriethoxy silane (APTS), Cot/DAP-urea-APTS, showed the highest amount of flame-retardant absorbed on the fabric during padding (24.5 wt%) with 9.9 wt% retained after soaking. For samples exposed to plasma after pad-drying, after water-soaking there is clear evidence of flame-retardant retention having increased compared to respective samples not exposed to plasma in spite of the relatively large errors listed in Table 1. These latter are particularly high for the lowest retention values but lower for the higher values, which are the more important ones under consideration here. Since Cot/GM-APTS_PL(N_2_/O_2_) and Cot/DAP-urea-APTS_PL(N_2_/O_2_) samples showed significant increases in FR retention after water-soaking, these were characterized more fully together with their respective controls by TGA/DTG, SEM/EDX and for flammability.

#### 2.1.2. Effect of Plasma as Pre- and Post-Treatments on Fabrics Spray-Impregnated with DBAP

In these experiments, because both the plasma power was double that in the previous work (i.e., 200 W/m^2^·min) (see Section 4.3) and mass losses of the cotton control after single exposures were recorded and listed in Table 2. This data shows that the mass of cotton fabric has decreased after plasma/UV treatment significantly by values higher than that seen in Table 1 for the unexposed (4%) and subsequently plasma/UV exposed (6%) cotton control after water-soaking at a plasma power of 200 W/m^2^·min. These higher mass losses may be explained in terms of fibre surface ablation as explained in our earlier publication [18]. The oxidative role of carbon dioxide was also noted when mixed with either nitrogen or argon relative to the lesser effects of exposing cotton under 100% argon alone. This effect was suggested to be a consequence of its conversion to CO, O_2_, and O_3_, formed via plasma-derived UV photolysis [34]. The presence of the laser was found to be associated with lower formation of radical species probably via their termination or removal from fibre surfaces [18]. Radical formation was considered to be the precursor of low molecular weight, surface cellulose degradation products. This would explain the lower mass losses when the combination of plasma and UV laser was used.

Thus the mass loss after plasma exposure is greater than that due to water-soaking alone. Surprisingly perhaps, mass loss after plasma exposure alone without the UV laser (samples with suffix P) is greater than that with UV (samples with suffix PL). This order of loss appears to be independent of the plasma gas used but mass loss is greatest for plasma alone in the presence of 20% carbon dioxide. The mass loss reduces when argon is the inert gas and lowest when 100% argon is present whether UV is present or not. Again, the particular reactivity of the Cot_PL(N_2_/CO_2_) mixture was noted in our previous publication [18].

Table 3 presents the percentage add-on values for the applied DBAP (Fyrol 6) flame retardant before and after water soaking for unexposed and exposed samples and the corresponding percentage retentions of FR.

It is evident that while a residual amount of DBAP exists on the unexposed, impregnated cotton sample (Cot/DBAP) after water-soaking, for exposed fabrics after soaking, the application of plasma, both with and without laser, increases the retention of DBAP. The samples with single pre-exposures with laser of the Cot_PL(Ar/CO_2_)_DBAP and those experiencing post-exposures (Cot/DBAP_PL(N_2_/CO_2_)) show retentions in excess of 10%. That the pre- and post-exposure with laser treatment of the Cot_PL(Ar/CO_2_)_DBAP_PL(N_2_/O_2_) sample gave a reduced retention of 5% suggests that the second exposure removes significant amounts of the fire retardant initially deposited. This low value may be reflected in the low initial add-on of 22 wt%, which is up to two thirds of values for the pre-exposed fabric samples although the Cot_PL(Ar/CO_2_)_DBAP (with laser) sample has the same low add-on. The role of the UV laser appears to be one of increasing the retention values of these latter samples.

### 2.2. Scanning Electron Microscopy/Energy Dispersive X-ray Spectroscopy (SEM/EDX) Results

Figure 1 shows SEM images for cotton fibres coated with guanidine monophosphate, GM. Compared to the SEM image of pure cotton in Figure 1a, fibres in Figure 1b appear to be covered with the flame-retardant and the surface texture or roughness has increased. However, after plasma treatment Figure 1c shows less FR (GM) coverage and the fibres look smoother. These results suggest that the plasma/UV energy is sufficient to etch away some of the surface flame retardant, hence leading to a decrease in the dry add-on value in Table 2.

After introducing GM and APTS to the fabric (sample Cot/GM-APTS) as shown in Figure 2a, the flame-retardant appears to be evenly spread on the fibre surfaces and after plasma/UV treatment, (Cot/GM-APTS_PL(N_2_/O_2_)) fibre surfaces in Figure 2c appear to be similar, which suggests that the coating was either not or little affected by the plasma/UV exposure. Meanwhile, after water-soaking, the unexposed Cot/GM-APTS sample fibre surface image in Figure 2b is smoother and indicates that most of the flame-retardant and silicon-based layer has been washed out as suggested by the percentage FR retention values in Table 1. After water-soaking the plasma/UV-treated Cot/GM-APTS_PL(N_2_/O_2_) sample fibre surfaces in Figure 2d appear very much like those of pure cotton (Figure 1a) although as Table 1 shows, there is a residual percentage of 5.2% present.

Since fibre surfaces showed little or no evidence of surface flame retardant aggregations, EDX was undertaken to observe the presence of respective species possibly embedded in or absorbed by cotton fibre surfaces which would reflect the add-ons listed in Table 1. EDX results shown in Figure 3 and Table 4 give measures of the average weight percent of the elements present on cotton fibre surfaces for Cot/GM and Cot/GM-APTS samples before and after plasma/UV exposure. The EDX results generally support the FR retention percentages calculated in Table 1 in terms of the relative percentages of the elements N, Si and P reflecting those values of the former.

The errors in values listed in Table 4 show that they are <±10% for the major elements C and O, but much higher for the minor elements present, namely N, Si and P. However, considering the respective magnitudes of these, plasma/UV laser exposure of Cot/GM has decreased the phosphorus percentage weight from 4% before the treatment to 1% after the treatment. While the weight percentages of carbon and oxygen have increased slightly after plasma/UV treatment to 47% and 48%, respectively, although within error these may be considered to be the same. Assuming they are valid, these value changes may be attributed to probable surface removal of flame retardant species as evidenced by respective reduction in phosphorus percentage although the value for nitrogen is unchanged.

Introducing APTS to the Cot/GM samples has reduced the percentage of surface fibre carbon significantly as shown in Figure 3c,d from 44% to 34% (Cot/GM-APTS) and from 46% to 31% (Cot/GM-APTS_PL(N_2_/O_2_)), respectively (see Table 4). Additionally, the results show that APTS has introduced to the cotton fibre surfaces 4 wt% silicon. In addition, there is an increase in the percentage of nitrogen from 5% for Cot/GM to 13% for Cot/GM-APTS and after plasma treatment to 16% for the Cot/GM-APTS_PL(N_2_/O_2_) sample arising from the nitrogen species in both the GM and APTS chemical structures. Again, this supports the SEM image, Figure 2c, where the GM-APTS surface layer on the fibres was little affected by plasma/UV treatment.

After water-soaking the surface carbon percentage of Cot/GM-APTS_PL(N_2_/O_2_) has increased slightly, due to washing out of the flame retardant, although oxygen levels are little changed. The weight percentage of silicon has reduced for the exposed Cot/GM-APTS_PL(N_2_/O_2_) sample to 1%. In-addition, significant reductions in the weight percentages of nitrogen-containing species following plasma exposure in both unexposed and plasma-exposed Cot/GM-APTS samples have occurred after water-soaking and the Cot/GM-APTS_PL(N_2_/O_2_) sample has still retained about 1 wt%. Phosphorus levels, however, while having reduced following water-soaking from 5 wt% for both unexposed and exposed samples to about 2 wt% levels. Taking into account the errors within the percentage values in Table 4, it would appear that loss of silicon species is greater than that of N- and P- containing entities within the combined GM-APTS formulation.

### 2.3. X-ray Photoelectron Spectroscopy (XPS) Analysis

#### 2.3.1. Fabrics Impregnated with FRs Comprising Combinations of DAP, Urea, TEOS, GM and APTS

XPS was used to analyze the elemental concentrations on the surfaces of Cot/DAP-urea-APTS samples with and without plasma/UV treatment and after water-soaking. The results are shown in Table 5 and Figure 4 and Appendix A as well as those for pure cotton as a control. Figure 4 shows the basic XPS spectra and Appendix A, the derived high resolution spectra C(1s), O(1s), N(1s) and Si(2p) components. For most elements in Table 5, moderate to strong photoelectron cross-sections, ensures that errors maybe less than 1% concentration. This is a minor amount if it is a major element present at the surface >10% such as carbon and oxygen, but becomes a proportionally larger error as the amount decreases especially when present at only a few % (N, Si, P).

For the pure cotton fabric sample, Cot (unexposed to plasma/UV and without water-soaking), spectra were only detected of carbon and oxygen elements on the surface in addition to 1% nitrogen most likely resulting from impurities present in the commercial fabric, possibly from contaminating crease-resist finish residues often present in the industrial processing environment. Conversely, in flame-retardant samples, both with and without plasma/UV treatment, nitrogen, silicon and phosphorus elements were detected in significant concentrations as noted also following EDX analysis (see Table 4). After water-soaking, the concentrations of carbon C(1s) have decreased, although the oxygen and silicon concentrations on the surface have increased. The nitrogen and phosphorus concentrations of flame-retardant samples were almost the same after water-soaking and the apparent slight increase in phosphorus concentration after plasma treatment could be within the experimental error. A more full discussion of the interpretation of this data is presented in Section 3.1.

O/C, N/C and Si/C atomic ratios have slightly increased after plasma/UV laser treatment compared to those with no plasma treatment. As a result, it may be concluded that N_2_/O_2_ plasma treatment has increased the percentage of oxygen-containing species at fibre surfaces. In addition, as the N/C ratio has increased in both samples compared to pure cotton, it can be concluded that nitrogen from the DAP and urea was retained on the fibre surfaces. Moreover, the Si/C ratio in the Cot/DAP-urea-APTS_PL(N_2_/O_2_) sample after water-soaking has increased to 0.1% compared to the unexposed analogue exposure (0.06%) and so it may be concluded that the APTS coating has interacted and possibly cross-linked on to the cotton cellulosic chains on fibre surfaces.

Bearing in mind the errors listed in Table 5 which shows them to increase as elemental content decreases, Table 6 shows that further analysis of the C(1s) data for these flame-retardant samples with and without plasma/UV post-treatment, indicates a significant increase in the C-C and C-H bond concentrations with the additional possible formation of C-N bonds, relative to pure cotton [35,36]. After plasma/UV treatment, the overall C-O bond percentage has increased compared to that of the non-plasma-exposed sample. However, analysis of the O(1s) data, suggests that the oxygen species do not show any significant change in the C-O-C bond or C=O bond concentrations. Si-O species have increased after plasma treatment in agreement with the increase shown in Table 5. In addition, and based on literature, the C-O-C bond may have been replaced by a Si-OH bond after introduction of the APTS flame-retardant [37].

In spite of the low Si(2p) concentrations in Table 5 and associated large errors, the resolved data in Table 6, show three components appearing in the spectrum at binding energies around 101, 102, and 104 eV corresponding to Si-N, SiO_2_, and siloxane (Si-O-Si) bonds, respectively and a fourth peak appears after plasma treatment/UV exposure at 103 eV corresponding to silicate (SiO_3_)^2−^. In addition, the SiO_2_ and Si-N concentrations appear to have decreased and the siloxane concentration to have increased significantly. Both decreases may relate to surface oxidation after N_2_-O_2_ plasma gas treatment where SiO_2_ and Si-N were replaced by Si-O species including silicate. Furthermore, there is the chance that APTS present after water-soaking has fully cross-linked to form a silica network including siloxane (Si-O-Si) groups on the surface [38,39].

Table 6 also shows the N(1s) resolved data, which although again associated with large error, suggests that where the N-O species have increased after plasma/UV treatment, this is accompanied by decreases in the amide (urea) and ammonium species (DAP) on the surface compared to non-plasma-treated samples [40,41,42]. In addition, after plasma exposure a new peak has appeared with bond energy 401.69 eV that may represent the imide (C=O)-N-(C=O) group. In the literature it is reported that the N(1s) peak mainly represents primary or secondary amine groups (-NH-, -NH_2_) and appears at around 399.36 eV and for cationic amino groups with a binding energy of 401.09 eV [43].

#### 2.3.2. Fabrics Spray-Impregnated with DBAP

XPS analysis of the elemental concentrations on the surface of control, Cot/DBAP, Cot/DBAP_PL(Ar/CO_2_) and Cot_PL(Ar/CO_2_)_DBAP_PL(N_2_/O_2_) fabric samples after water-soaking are presented in Appendix A with percentage contents and ratios listed in Table 7. Again, as noted previously, errors in respective element concentrations again are less when the latter are large and so those associated with nitrogen and phosphorus are high. Pure cotton fabric spectra only detected carbon and oxygen elements as expected. Plasma-exposed samples containing DBAP in particular showed reductions in surface carbon concentrations as might be expected arising as a consequence of the residues of DBAP still present, which is evidenced by the low percentage levels of phosphorus present. Of interest also is that low percentages of nitrogen (as N(1s)) were detected, although the same level is present in the control as noted also above.

Plasma/UV laser-post-exposed samples, under N_2_/CO_2_ plasma gas have marginally increased phosphorus element concentrations compared to Cot/DBAP, although experimental error most likely suggests little change. Notably, the effect of a post-plasma treatment in the Cot_PL(Ar/CO_2_)_DBAP_PL(N_2_/O_2_) sample indicates a reduction in both N and P percentages, most likely the consequence of partial removal of DBAP.

Table 7 also shows that the O/C ratio has increased after plasma/UV treatment significantly, with post-plasma exposure (Cot/DBAP_PL(N_2_/CO_2_) giving the highest ratio of 0.6, again reflecting the highest residual add-on percentage (13%) in Table 3.

With regard to the further analysis of the C(1s) and O(1s) data, Table 7 and Appendix A show that plasma/UV-treated samples indicate a significant decrease in the C-C and C-H bonds and N species and an increase in the total C-O species concentrations. O(1s) components for the three samples comprise two peaks corresponding to C-O-C or C-O-P or P-O-P and C=O, P=O bonds at bond energies around 532.9 and 530.8 eV, respectively [35,36,37,38]. After plasma/UV pre-treatment under the Cot/DBAP_PL(N_2_/CO_2_) condition, there occurs a decrease in the C-O-C/C-O-P/P-O-P bond concentrations, while the C=O and P=O concentrations have increased to 11%. However, the Cot_PL(Ar/CO_2_)_DBAP_PL(N_2_/O_2_) sample, shows a reduction in the C=O, P=O bond concentrations, confirming that the post-plasma treatment has been removed some surface flame retardant as reflected in the lower residual add-on in Table 3. Assuming the latter to be the case, the increase in the C-O-C/C-O-P/P-O-P concentration is most probably the consequence of an increase in the surface C-O-C bond concentration as a consequence of cellulose chains being revealed as surface DBAP (and hence C-O-P bond concentration) is etched away during the post-plasma/UV exposure to the N_2_-O_2_ oxidative atmosphere.

### 2.4. Thermogravimetric Analysis (TGA/DTG/DTA) in Air

#### 2.4.1. Cotton after Water-Soaking with and without Plasma/UV Treatment (No Flame-Retardant)

The differences in the thermal behaviour under air for unexposed cotton fabric before and after water-soaking is significant as shown in Figure 5. The temperature for 10% mass loss, T_10%_, has shifted from 290 °C for the non-water-soaked (Cot) sample to the higher temperature of 313 °C for the water-soaked sample (Cot-(ws)), with an increase in the temperature of rate of maximum mass loss (DTG2) shifting from 461 for the unsoaked sample to 469 °C for the water-soaked sample (see Table 8). These differences may relate to some inorganic species (such as metal salts) present in the fabric after the bleaching and finishing manufacturing stages, which promote cellulose degradation and char formation.

Furthermore, plasma/UV-exposed cotton fabric after water-soaking (sample Cot_PL(N_2_/O_2_)-(ws), Figure 5), reduces T_10%_ from 313 to 307 °C, suggesting slight sensitisation to thermal oxidative pyrolysis. However, plasma exposure appears to have little effect on the subsequent DTG peak temperatures, although the amount of char at 400 °C and above has slightly increased.

#### 2.4.2. Fabrics Impregnated with FRs Comprising Combinations of DAP, Urea, TEOS, GM, APTS and DBAP

TGA/DTG curves in air are shown in Figure 6 for the applied APTS-GM formulations before and after water-soaking and these typify all applied formulations listed (see also Appendix A for respective TGA/DTG responses for water-soaked Cot and Cot/DAP-urea-APTS samples).Table 8 shows all derived data with significant differences in their values and in particular the decomposition temperature, T_10%_, relative and char residue values at 400 °C relative to the cotton control, Cot. The shifts in TGA curves and derived data both before and after water-soaking reflect the relative initial add-ons and retentions listed in Table 1 and Table 3. For example, the Cot/DAP-urea-APTS sample before water-soaking has the lowest T_10%_ value of 270 °C, which might be expected for an efficient condensed phase flame retardant [44] rising to 287 °C after water-soaking reflecting the 9.9% retention of applied FR (Table 1). This same sample with a plasma/UV post-treatment (Cot/DAP-urea-APTS_PL(N_2_/O_2_)-(ws) in Table 8) has an increased retention of 12.1% after water-soaking, and a slightly decreased T_10%_ = 285 °C reflecting this former value. However, the highest level of flame retardant retention for pre-exposed fabric containing DBAP (11.0%, Table 5) after water-soaking is the Cot_PL(Ar/CO_2_)_DBAP-(ws) sample and this has an only slightly T10% = 300 °C relative to pure cotton (300 °C), perhaps reflecting the absence of significant nitrogen-phosphorus synergy within this sample [44]. Char levels at 400 °C are 33.0 and 16.4%, respectively, corresponding to their respective retention levels and some level of condensed phase activities.

Table 8 also indicates that DTG1 peaks, measures of the maximum rates of decomposition temperatures, have all reduced in the presence of applied flame retardants with in the case of the GM-, DAP-, urea- and APTS-containing formulations little dependence upon whether or not samples have been soaked. For DBAP-containing samples, shifts to lower temperature are extremely sensitive to water-soaking with a DTG1 value for the water-soaked Cot/DBAP-(ws) sample (332 °C) approaching the value for the control of 339 °C. These higher temperature shifts are in spite of the relatively high FR retention percentages after soaking observed in Table 3.

DTG2 values, indicating maximum rates of char oxidation are all higher than that for the control (469 °C) when in the presence of flame retardant, which are evidence of their greater resistance to oxidation. With regard to the presence of guanidine monophosphate/aminopropyltriethoxy silane (GM-APTS) combinations, the DTG2 values span the range 486–490 °C and so are little influenced by either plasma/UV post-treatment or water-soaking. However, the presence of diammonium phosphate/urea/aminopropyltriethoxy silane (DAP-urea-APTS) combinations exhibit a wider range from 489–524 °C with the unsoaked and both untreated and plasma-treated samples showing the highest values (524 and 511 °C, respectively). After water-soaking, both unexposed and plasma-exposed samples have similarly decreased values (489 and 490 °C, respectively). This suggests that the more soluble DAP and urea components are most likely responsible for the initial higher char oxidation resistance and that their water solubility is minimally plasma/UV exposure and so any increased FR retention and increased DTG2 values are a consequence of the APTS which has formed some form of physicochemical bonding to the underlying cellulose chains.

Samples containing DBAP, while showing a DTG2 range 495–504 °C, for all unsoaked samples, also show much reduced values (472–490 °C) after water-soaking. While the water-soaked, unexposed Cot/DBAP sample shows a reduction to 476 °C reflecting its low after-soaking water retention value of 2.1% (Table 3), the samples Cot_PL(Ar/CO_2_)_DBAP-(ws) and Cot/DBAP_PL(N_2_/CO_2_)-(ws), both have higher and similar DTG2 values of 490 and 489 °C correlating well with their after-soaking retentions of 11 and 13%, respectively. However, while the addition of a post-plasma/UV exposure to the Cot_PL(Ar/CO_2_)_DBAP sample (Cot_PL(Ar/CO_2_)_DBAP_PL(N_2_/O_2_)-(ws)) reduces the FR retention from 11 to 5%, the DTG2 value remained at 489 °C. The role of the UV laser is not clear since while its presence increases FR retention values for both samples to these higher levels, in the absence of UV laser the reduced retentions of pre-exposed Cot_PL(Ar/CO_2_)_DBAP-(ws) to 7% and post-exposed Cot_PL(Ar/CO_2_)_DBAP_PL(N_2_/O_2_)-(ws) to 10% are matched by respective reduced DTG2 values of 472 and 490 °C. The former lower value suggests a minimal effect of FR on char stability perhaps reflecting the lower after-soaking FR retention of 7% (see Table 3).

The importance of the DTG2 values above as indicators of retained flame retardant effectiveness may be examined further by relating them to the TGA-derived char levels at 400 °C also listed in Table 8 and plotted in Figure 7. This suggests that the resistance of char to oxidation correlates well with the char residue, which is a consequence of the condensed phase activity of DBAP.

### 2.5. Flammability Testing

While the cotton control samples with and without plasma treatment burned completely after removing the flame as expected, as shown in Figure 8 for unsoaked sample examples from Table 9 (for Cot/GM, Cot/GM_PL(N_2_/O_2_), Cot/GM-APTS, Cot/GM-APTS_PL(N_2_/O_2_), Cot/DAP-urea-APTS and Cot/DAP-urea-APTS_PL(N_2_/O_2_) samples) for both unexposed and plasma/UV exposed conditions, the presence of each flame-retardant formulation influences the burning behaviour based on its type and that subsequent plasma/UV treatment has only a secondary influence. In most cases, extinction of the ignited fabric occurred before flame reached the top of the sample. After water-soaking, however, the burning test results showed that all samples burnt along their entire lengths, although heavy chars were left (see Figure 9). Similar results were recorded for DBAP-containing samples (see Appendix A) and the results for all fabrics tested in terms of damaged lengths and burning rates were determined and are listed in Table 9, both before and after water-soaking.

It is evident that before water-soaking the presence of GM alone applied at a nominal 3% phosphorus level was insufficient to generate a high level of flame retardancy, although it does promote charring. Only the combinations GM-APTS and DAP-urea-APTS applied to cotton have generated self-extinguishing properties with the latter formulation giving the lowest damaged lengths. Closer inspection shows that plasma/UV post-treatment has caused a reduction in the damaged length and a decrease in the burning rate for the Cot/GM-APTS_PL(N_2_/CO_2_) and Cot/DAP-urea-APTS_PL(N_2_/CO_2_) samples relative to their respective unexposed analogues.

The behaviours of the cotton/FR samples after water-soaking are very different from the respective unsoaked samples (Figure 7) and no samples now self-extinguish although heavy chars are left reflecting the post-water-soaking FR retention values in Table 1. This is especially the case for the water-soaked Cot/DAP-urea-APTS-(ws) and Cot/DAP-urea-APTS_PL(N_2_/O_2_)-(ws) samples which show the apparently most dense of chars, which also correlates well with these samples having the highest post-soaking TGA 400 °C char values of 33.0 and 31.0%, respectively (see Table 8).

Similar behaviour was seen for the DBAP-containing fabrics in that once again and in spite of the FR retention percentages in Table 3, the water-soaked samples unlike their unsoaked analogues did not self-extinguish. However, the dense chars observed in Appendix A were not observed in the water-soaked samples in Appendix A, reflecting their lower TGA 400 °C char levels in Table 8 compared with that from the analogous Cot/DAP-urea-APTS_PL(N_2_/O_2_) samples. This perhaps reflects the expected condensed phase, char-forming activity of the latter [44] compared with some possible gas phase activity and hence lower char-forming tendency of DBAP. To better distinguish the changes in flame retardant activity, LOI values were also recorded and Table 9 shows that the initially high values (>27.0 vol%) before soaking, reduced significantly to 19.5–20.0 vol% after water-soaking against a control value of 18.5 vol%. The slight increases in LOI values of 1–1.5 vol%, while not being large, reflects the respective FR retention percentages in Table 5 and the increased TGA chars and DTG2 temperatures (Table 8 and Figure 7). The effect of post-plasma treatments have appeared to generate the highest LOI values with the Cot/DBAP_PL(N_2_/CO_2_) sample achieving a value of LOI = 20.0 vol% with FR retentions of 10–13% with no evident effect of the absence or presence of UV laser in terms of LOI and DTG2 values.

### 2.6. Analyses of Chars from Vertical Strip Test Residues

In order to more directly observe whether there was still residual flame retarding elements remaining after water-soaking the SEM/EDX behaviours of selected chars obtained after vertical strip testing was investigated. Based on their relatively high FR retention values in Table 1, Cot/DAP-urea-APTS samples with and without plasma/UV treatment were selected. The water-soaked samples yielded the apparently most dense chars observed in Figure 8f and Figure 9e and so lent themsleves to more facile analysis.

Comparison between the two SEM char images in Figure 10 shows that for Cot/DAP-urea-APTS-(ws) (Figure 10a), the fibre chars are separated reflecting the parent fibres, while for Cot/DAP-urea-APTS_PL(N_2_/O_2_)-(ws) (Figure 9b), after plasma/UV exposure, fibre chars are less distinct with some additional covering with a thick layer of char. The related EDX spectra are shown in Figure 11. The weight percentages for the elements in each char found by the EDX analysis are presented as averages in Table 10 with respective errors expressed after following analysis of 5 different areas of each sample.

While the errors in the values in Table 10 are <±10% for the major elements C and O, they increase for the minor elements present (N, Si, P) and are >±100% for silicon. Notwithstanding these, the weight percent of nitrogen-containing species appears to be slightly higher in the unexposed Cot/DAP-urea-APTS-(ws) sample than the exposed analogue suggesting that subsequent sample plasma/UV exposure may have preferentially removed some of the urea component. The increases in the weight percentages of silicon and phosphorus in the plasma-exposed Cot/DAP-urea-APTS_PL(N_2_/O_2_)-(ws) sample suggest that concentrations of these elements via possible bonding of the APTS and DAP components to cotton fibre surfaces has occurred. These results corroborate the relative char lengths and burn rates shown in Table 9, which are both reduced after plasma/UV exposure.

## 3. Discussion of Possible Plasma-Induced, FR-Cellulose Bonding

### 3.1. The Role of Silanes (TEOS and APTS) in Promoting FR-Cellulose Bonding

It is evident from our previous work regarding the action of combined plasma (200 W/m^2^·min)/UV exposure on cotton [18], which showed that in the presence of 20% argon or 20% carbon dioxide/80% nitrogen atmospheres promotes generation of radicals and addition of 20% oxygen the formation of –CO.OH groups, fibre surface reactivities have increased. Similar reactions may also occur during the plasma/UV exposure of flame retardants as well.

As Table 1 has shown, the high post-soaking, residual flame retardant levels exhibited in the post-exposed samples Cot/GM-TEOS_PL(N_2_/O_2_) (4.6%), Cot/GM-APTS_PL(N_2_/O_2_) (5.2%) and Cot/DAP-urea-APTS_PL(N_2_/O_2_) (12.1%), which are significantly greater than their unexposed analogues, suggests that some degree of FR-cellulose chemical bond formation has been introduced. That the Cot/DAP-urea-TEOS_PL(N_2_/O_2_) sample has a lower (3.9%) retention than this last sample, indicates that the presence of APTS is more beneficial towards such bond formation than is TEOS, although the FR retentions of the Cot/GM-TEOS_PL(N_2_/O_2_)and Cot/GM-APTS_PL(N_2_/O_2_) samples are much more similar and, within experimental error, most likely the same. It is also noteworthy that the Cot/DAP-urea sample after plasma/UV exposure rises from the low level retention level of 0.4% (as expected for such a water soluble flame retardant combination) to 2.3% after exposure and then to 3.9% following addition of TEOS in sample Cot/DAP-urea-TEOS_PL(N_2_/O_2_). Similarly, the slightly less water-soluble guanidine monophosphate in the unexposed Cot/GM sample shows a retention value rising from 1.8% to 3.3% after plasma/UV exposure and the addition of TEOS increased it again to the 4.6% level. It may be concluded, therefore, that both TEOS and APTS have increased any potential FR-cellulose chemical bonding effect with the latter yielding the greater. In the case of DBAP-containing sample FR retentions in Table 3, it is evident that pre-plasma/UV exposed cotton fabrics under an Ar-CO_2_ plasma mixture have also been activated sufficiently to form some chemical bonding with cellulose and that a single post-exposure under a N_2_-CO_2_ plasma atmosphere has produced a similar effect. However, post-exposure under the N_2_-O_2_ oxidising atmosphere following a pre-Ar-CO_2_ plasma mixture exposure in sample Cot_PL(Ar/CO_2_)_DBAP_PL(N_2_/O_2_) has removed some of the bonds formed following pre-exposure condition.

The possible nature of any FR-cellulose bonding occurring during plasma/UV exposure of FR-impregnated samples may be discussed with regard to the XPS data in Table 5 and Table 6 by referring to the Cot/DAP-urea-APTS samples in the first instance. These, while indicating little change in nitrogen presence, show a higher retention of silicon and phosphorus after plasma/UV exposure with the presence of Si-N, siloxane Si-O and SiO_3_^2−^ species featuring as well as formation of insoluble SiO_2_. The concentration of siloxane groups in Table 6 as a percentage of the total Si(2p) presence is particularly high at 50%. The low levels of phosphorus observed in Table 5 suggest that most P-containing moieties were water soluble and so had little if any presence in cross-linking reactions with cellulose. However, EDX data from respective chars of water-soaked Cot/APTS-DAP-urea-(ws) and Cot/DAP-urea-APTS_PL(N_2_/O_2_)-(ws) samples in Table 10 show significant increases in concentrations of silicon and especially phosphorus in the latter after burning, while again nitrogen levels remained almost unchanged within error. Assuming that the retained flame retardant components were functioning in the condensed phase, these results further suggest efficient bonding between silicon and cellulose or via a bridging nitrogen-containing species. The slight apparent increased presence of phosphorus and its associated error with respect to the XPS analysis data for the Cot/DAP-urea-APTS_PL(N_2_/O_2_) sample in Table 5 are perhaps a consequence of the surface fibre phosphorus concentrations being too low for accurate XPS analysis. It is well known that urea increases the ease of penetration of many chemical species into cotton fibres [45] and that such urea-cellulose hydrogen bonding will also act as attractive sites for phosphate ions released by DAP. Thus these low surface P levels are most likely because of preferential absorption of the DAP-urea synergistic components within the fibre [30,46,47,48], but which are subsequently revealed in greater concentration within the char after burning (Table 10).

Thus in conclusion and bearing in mind the errors in the XPS and EDX data, it would appear that the formation of Si-O and to a lesser extent Si-N bonds most likely comprise the majority of possible bridging bonds between APTS-derived siloxane moieties and cellulose (as (APTS)Si-O-cellulose links) and bridging via the NH_2_ group in APTS as cellulose-NH(APTS)SiO-cellulose bonds. It might be added at this point, that unpublished infrared spectroscopy studies of chars showed the presence of Si-O-Si bonds in both Cot/DAP-urea-APTS-(ws) and Cot/DAP-urea-APTS_PL(N_2_/O_2_)-(ws) samples, but whether or not there was a significant the difference in their intensities between unexposed and exposed sample was inconclusive [49]. Nevertheless, this proposal could also explain why the post-water-soaking retention of the Cot/DAP-urea-TEOS_PL(N_2_/O_2_)-(ws) sample at 3.9% is much less than for the Cot/DAP-urea-APTS_PL(N_2_/O_2_)-(ws) sample at 12.1% (see Table 3). Similarly, the retention for the Cot/GM-TEOS_PL(N_2_/O_2_)-(ws) sample at 4.6% is less than that for the Cot/GM-APTS_PL(N_2_/O_2_)-(ws) sample at 5.2%, although the difference is much less.

### 3.2. Potential FR-Bonding in DBAP-Impregnated Cotton

With regard to the effect of pre- and post-plasma/UV exposure of fabrics to which diethyl N, N bis (2-hydroxyethyl) aminomethylphosphonate (DBAP) has been applied gave rise to similar post-water-soaking retention levels for both Cot/Ar-CO_2_/DBAP (11.0%) and Cot/DBAP_PL(N_2_/CO_2_)-(ws) (13.0%) (see also Table 5). These results indicate that both pre- and post-plasma/UV exposure conditions lead to some form of bonding between the DBAP and cellulose. Furthermore, the presence of laser radiation appears to have increased DBAP retention relative to its absence for both pre- and post-exposure conditions. Previous work on pure cotton showed that free radical production was greater when no UV laser was present suggesting that its presence caused their deactivation by recombination of other reactions [18], which might suggest that any DBAP-cellulose bonding was not dependent on significant cellulose radical formation. However, these earlier studies do not preclude the formation of active radicals during plasma/UV exposure of DBAP present on cotton. It is also interesting to note that while a single plasma/UV exposure either pre- or post-FR impregnation, increased DBAP retention, the application of a second exposure (Cot_PL(Ar/CO_2_)_DBAP_PL(N_2_/O_2_)) significantly reduced it to 5%, suggesting that removal of surface attached flame retardant has now occurred.

The XPS data in Table 7, as stated above in Section 2.3.2, indicates that the post-plasma/UV treatment of DBAP-impregnated cotton has promoted an apparent decrease in the C-O-C/C-O-P/P-O-P bond concentrations and an increase in the C=O and P=O concentrations, while the total phosphorus content after water-soaking has increased from 0.7 to 1.0% with respect to the unexposed sample. No change in the concentration of nitrogen was observed in terms of C-N bond presence. The doubly plasma/UV-exposed sample Cot_PL(Ar/CO_2_)_DBAP_PL(N_2_/O_2_) XPS data showed both reductions in surface phosphorus and nitrogen concentrations, confirming that indeed loss of some surface-bonded DBAP has occurred as reflected in its reduced FR retention value and TGA residue at 400 °C in Table 8. As stated above (Section 2.3.2) and bearing in mind the associated errors, the apparent increased XPS-derived concentrations of the C-O-C and C-O-P bonds and decrease in C=O and P=O signals could also be interpreted as removal of some surface-bonded DBAP since with regard to the former, the relative concentration of C-O-C bonds in underlying cellulose chains would increase as indicated in the slight increase in all C-O bonds from 73 to 76% of all C(1s) species. Burning rate and LOI values, however, are little affected by the second post-plasma/UV exposure (Table 9).

If any DBAP-cellulose bonding had indeed occurred, then within the error of the XPS data, it can be inferred that it occurs via the phosphonate entity present. If cellulose bonding had occurred via the pendant -N.CH_2_.CH_2_-OH groups in DBAP, an increase in bonded nitrogen would have been expected after plasma-exposure, a fact not present in Table 7. Given that the both C=O and P=O concentrations have increased, the application of the N_2_-CO_2_ post-impregnation exposure appears to have promoted some degree of oxidation most likely in cellulose chains (forming C=O). Furthermore, the increase in surface P=O concentration suggests its formation at a DBAP- cellulose bonding site. This in turn would reduce surface cellulose and hence C-O-C concentrations, which would explain the reduction in the combined C-O-C and C-O-P bond signals. DBAP-cellulose bond formation could occur via an effective interchange reaction such as Reaction (i) in Figure 1.

However, since there is no distinction between C-O-C and C-O-P in the data in Table 7, it is also possible that the overall reduction in their concentration is borne also by the C-O-P group as a consequence of C-(PO) bond scission at the DBAP phosphonate moiety as a consequence of its much lower bond energy (~264 kJ/mol) compared with that of the adjacent (PO)-O bond (~350 kJ/mol), as shown by Reaction (ii). This would yield two free radicals which could then attack adjacent cellulose although the unchanging nitrogen level after plasma treatment suggests that the first radical presence may not be favoured and the second radical would restore the P-O bond concentration. In addition there is the possible direct formation of a C-P bond between an anhydroglucopyranose ring carbon in cellulose and DBAP via an oxidative radically initiated reaction (iii), although this would perhaps be less favoured under the combined plasma/UV laser where lower radical formation has been observed [18].

While both these reactions are speculative and based principally on the XPS data and its associated errors, of the two, the more likely might be Reaction (i) with DBAP bonding to adjacent cellulose chains via (PO).O-cellulose bond formation.

## 4. Materials and Methods

### 4.1. Fabric and Preparation

Commercially bleached woven cotton fabric (Cot) with area density 251.5 g/m^2^ and typical of a workwear or lightweight furnishing fabric was used without further preparation. The fabric was sourced from Whaleys Ltd., Bradford, UK. Cotton fabric was cut into pieces measuring 25 cm × 100 cm ready for plasma treatment and FR coating.

### 4.2. Flame Retardant (FR) Selection and Application to Cotton

The following selection of non- or semi-durable flame retardant components based phosphorus-, nitrogen- and/or silicon species were selected based on their known effectiveness on cotton with structures listed in Figure 2 and sourced from Sigma Aldrich Ltd. (Irvine, North Ayrshire, UK): tetraethoxysilane (TEOS) 13% Si; diammonium phosphate (DAP) 23% P; urea; 3-aminopropyltriethoxy silane (APTS) 13%Si; and guanidine monophosphate (GM) 19% P. In addition, diethyl N, N bis (2-hydroxyethyl) aminomethylphosphonate (DBAP or Fyrol 6, 12%P) was provided by ICL Industrial Products (Tel-Aviv, Israel).

Other chemicals used were methanol of spectrophotometric grade, 99.9%, Triton-X 100 for use as a non-ionic wetting agent and hexamethyldisiloxane (HMDSO) also sourced from Sigma-Aldrich Ltd. (Irvine, North Ayrshire, UK), were used without further purification.

The flame retardant combinations of species in Figure 1 were based on the well-established, condensed phase-active guanidine monophosphate (GM) [48] or diammonium phosphate (DAP) in synergistic combination with urea [28,44] both alone and in combination with either tetraethoxysilane (TEOS) or 3-aminopropyltriethoxy silane (APTS). These two silanes were added in order promote silicaceous char formation [8] and also introduce the possibility of bonding to surface cotton fibre, cellulose molecules via formation of Si-O-O.cellulose bonds.

The FR solutions comprising combinations of DAP, urea, TEOS, GM and APTS (see Table 11) were prepared to achieve a nominal 3% phosphorus level in the application solutions which comprised combinations of various of the selected components combined also where relevant based on a molar P/N ratio of 5/1 and a molar P/Si ratio of 1/1 [12,28,48]. They were introduced to the fabric in a two-step procedure as follows:DAP-urea solution: 10 g of DAP and 2 g of urea were added to 90 mL distilled water. Then, the DAP/urea was introduced to the fabric by padding and dried in the oven at 105 °C for 5 min.DAP-urea-TEOS or -APTS samples: Either TEOS or APTS was dissolved in methanol within the fume cupboard. Then, dry samples already coated with DAP-urea were immersed in this solution and padded and dried in the oven at 80 ºC until completely dry.

Similar steps were followed for the preparation of guanidine monophosphate (GM) solution and GM-TEOS or -APTS samples.

The above various flame retardant formulations were applied using a laboratory padding method to achieve about 80% expression prior to drying in the open laboratory. Alternatively, for some experiments application by a laboratory spraying method was used.

Diethyl N, N bis (2-hydroxyethyl) aminomethylphosphonate (DBAP or Fyrol 6) (Figure 2f was selected for a second set of experiments because although it is was designed as a reactive flame retardant via its –CH_2_.CH_2_.OH groups for use in polyurethane foams [50], it comprises both phosphorus and nitrogen and so should be expected to act as a condensed phase FR on cotton. Furthermore, it is a water-insoluble, high boiling liquid (398.8 ± 27.0 °C at 760 mmHg), which after application to cotton may expected to partly volatilised during plasma exposure and so be activated for reaction with and grafting on to adjacent cellulose chains.

Because experiments using DBAP were undertaken at the MTIX Ltd., commercial establishment with limited experimental facilities, the formulation as a nominal 3% phosphorus solution was prepared in methanol (99.9% purity) in a 25/75 *v*/*v* ratio. This enabled rapid evaporation of the solvent thus enabling a spray method of application to be adopted. A laboratory pressure sprayer was used and adjusted to control the amount of flame retardant sprayed during coating in order to reach 100% wet add-on. In addition, trials were undertaken to achieve consistency of flame retardant application across each fabric specimen.

After introducing the flame retardant by padding or spraying, the wet and dry add-on % values were calculated following Equation (1). In particular, dry add-on values were calculated from cutting a square 10 cm × 10 cm from the treated fabric and dried for 5 min at 105 °C in a vacuum oven followed by 1 h at 80 °C in laboratory oven (W_1_) compared with a similar size of pure fabric (W_2_), which had been dried for 2 h at 80 °C following Equation (1).
Add-on % = 100 × (W_1_ − W_2_)/W_2_(1)

### 4.3. Atmospheric Plasma/UV Laser Treatment

The combined atmospheric plasma/UV laser system is a commercially available full-scale equipment manufactured by MTIX Ltd., Huddersfield UK, who define it as Multiplexed Laser Surface Enhancement (MLSE) atmospheric plasma/UV excimer laser source [22] and has been previously used in our previous studies and described elsewhere [10,18]. After application of a specified flame retardant formulation, samples were exposed to MLSE combined plasma/UV laser exposure under conditions summarised in Table 11. The plasma gases used were N_2_^80%^/CO_2_^20%^, N_2_^80%^/O_2_^20%^ and Ar^80%^/CO_2_^20%^ and the power dosages were set at either 100 or 200 W/m^2^.min depending on the type of flame retardant formulation present (see Table 11 and Table 12) [18]. Samples treated with 200 W/m^2^.min plasma power were exposed twice to a single run at 100 W/m^2^·min. Nitrogen, carbon dioxide and argon gases were of technical quality supplied by the sponsoring company.

The choice of gas mixtures used for the various pre-, post- and combined plasma/UV exposure conditions defined in Table 11 and Table 12 were based on previous work [18] which had shown that their generation of radicals and new reactive functional groups in cotton cellulose would facilitate possible chemical interactions between applied flame retardant and adjacent cellulose chains. Other plasma parameters were fixed as follows:Four head electrodes (2 heads/side)Process speed: 20 m/minLaser energy: 228 W. However, some samples were treated with MLSE plasma only without UV laser.Electrode gap: 1.5 mmGas flow: 28 L/m

The samples were exposed to several cycles under atmospheric plasma. The plasma power dosage was calculated following these two equations for this plasma treatment.
Power dosage (W/m^2^·min) = (No. of plasma cycles) × (Power/surface coverage)(2)

In the first set of experiments FR-containing fabrics comprising combinations of either GM or DAP-urea alone or in combination with either TEOS or APTS are listed in Table 11 and cotton fabrics pre-impregnated with each were subjected to a post-plasma/UV treatment in order to promote possible FR-cellulose bond formation.

Samples to which DBAP (as Fyrol 6) was applied were subjected plasma treatments under the following conditions listed in Table 12 in order to assess the effects of higher plasma power (200 W/m^2^·min) and different gas compositions, which were selected based on previous experience as outlined above [18]. The uses of pre-plasma/UV exposure of fabrics prior to and post–plasma/UV exposure after DBAP application were selected in order to assess which condition might produce the greatest increase in FR-cellulose bonding. Furthermore, the use of both pre- and post-exposures would establish whether there was further FR-bonding improvement.

### 4.4. Wash Durability Testing (BS5651:1978)

To study the durability of the flame-retardant samples, fabric samples (with and without plasma treatment) were soaked in distilled water (1:20 *w*/*v*) with Triton x-100 as non-ionic wetting agent (0.5 g/1000 mL) for 30 min at 40 °C according to the procedure in British Standard BS5651-1978 [33]. The mass loss of flame retardant after water-soaking was calculated using Equation 3 where, W_b_ is the weight of the fabric before water soaking and W_a_ represents the weight of the fabric after water soaking.
Mass-loss, % = 100 × (W_b_ − W_a_)/W_b_(3)

The flame-retardant remaining on the fabric after water-soaking (Ws) was calculated by subtracting Equation (3) from Equation (1) as demonstrated in Equation (4), where W_p_ represents the percent of impurities removed from the pure fabric after water-soaking. Each sample was tested as three replicates and results averaged.
FR remaining, % = [Dry add-on, %] − [Mass Loss, %] + W_p_(4)

Levels of acceptable flame retardant retention for acceptable flame retardancy of the resulting fabrics might be expected to be above 90% or so, although actual values would be expected to be dependent on the flame retardant used and the flammability test requirements (see Section 4.8).

### 4.5. Morphological Characterization

A Hitachi S-3400N Scanning Electron Microscope (Hitachi High-Tech Analytical Science Ltd., Abingdon, UK) was used to study the changes in the surface morphology of the coated fabric samples before and after plasma/UV laser exposures. All the samples were mounted on aluminium stubs using SEM conductive adhesive tape. Then, the samples were sputter-coated with a conductive gold layer using a Quorum Technologies SC7620 sputter coater before SEM analysis with a beam voltage over the range 2–5 kV.

### 4.6. Surface Chemistry Characterisation (X-ray Photoelectron Spectroscopy (XPS))

The XPS spectra were determined using a Kratos Axis Supra instrument at the EPSRC National Facility at Harwell, UK for XPS, operated by Cardiff University and UCL. Samples were mounted on to copper tape and exposed to a 1486.7 eV Al (mono) ray with 150 W power. Each sample was analysed three times from different positions. Data were provided as VAMAS files and analysed using CasaXPS demo software (J.T.Grant, Surface Analysis Consulting, Clearwater, FL, USA). The C–C component of the C1s signal at 285.0 eV was also used as the reference value for the binding energy scale.

### 4.7. Thermal Behaviour

Thermal degradative behaviours of cotton fabrics before and after plasma treatment were studied using an SDT 2960 Simultaneous DTA-TGA instrument (TA Instruments, Wilmslow, UK). Sample sizes were 5.0–6.0 mg for cotton, the heating rate was 10 °C/min under air with a flow rate 110 mL/min and each sample was heated from 30 °C to 700 °C. The results obtained from the mass loss curves were analysed to determine the onset temperature of sample decomposition at a defined percentage mass loss, maximum rate of mass loss temperatures and the percent of mass residue at various temperatures.

### 4.8. Flammability Testing

#### 4.8.1. Vertical Flammability Strip Testing

This test follows the British Standard BS-ISO 6940 [51]. This test is used to measure the ease of ignition of a vertically orientated textile fabric as a single layer. The sample (size 200 mm × 80 mm) is oriented vertically and subjected to a small flame of length 25 mm directed on the face of the fabric. The time allowed for ignition was 10 s. After 10 s the flame is turned off and the sample will either stop burning (self-extinguish) or continue burning. An arbitrary pass/fail criterion was defined such that burn times of ≤120 s and the flame not reaching the top of the sample yielded a “pass”. The burning rate was also calculated as explained in Equation (5) as the averaged result of three specimens per sample.
Burning rate (mm/s) = (Burning damage length (mm))/(Duration of after flame (s))(5)

#### 4.8.2. Limiting Oxygen Index (LOI)

LOI is one of the earliest established methods of quantifying flammability of a material in terms of a single number that relates to the concentration of oxygen required to sustain burning [52,53]. Samples were tested according to BS4589-2; 1999 in which a sample of 50 mm × 100 mm dimension is fixed in a gas chamber, through which a mixture of nitrogen and oxygen gases in adjustable volumes is introduced. The sample was then subjected to a “candle-like” gas flame from a small propane burner at the sample top edge. Three specimens per sample were tested and results averaged.

## 5. Conclusions

It has been shown that the application of plasma/UV, preferentially to cotton fabric, which has been pre-impregnated with flame retardants comprising combinations of phosphorus-, nitrogen- and silicon-containing species, may increase the water-soak durability of these previously water-soluble components. This provides evidence for the formation of covalent bonds between one or more of the components and the underlying cellulose chains. In the case of the DAP-urea-APTS combination, XPS evidence suggests that plasma/UV exposure promotes the formation of siloxane bonds between the silicon atom within the 3-aminopropyltriethoxy silane component and adjacent cellulose chains. On the other hand, when the single flame retardant diethyl N, N bis (2-hydroxyethyl) aminomethylphosphonate (DBAP) is present on the fabric, it has been proposed that it may bond to cellulose via formation of either a (PO).O.cellulose bond.

These results are in broad agreement with our earlier work [10], which showed that heavier cotton furnishing-grade fabrics with area densities >330 g/m^2^ impregnated with a proprietary non-durable FR, showed enhanced durability after plasma/UV treatment, although no in-depth analysis of the mechanism involved was undertaken.

While both these flame retardant systems have retained post-water-soaking levels of slightly greater than 10%, these concentrations are insufficient to promote self-extinction in a simple vertical strip test, although significant increases in char formation both during such testing and also following TGA were observed. However, notwithstanding these low levels achieved, we have shown that plasma exposure may induce flame retardant bonding to cellulose, which if more fully understood, could give rise to significantly improved durability and hence flame retardancy levels sufficient to pass standard testing procedures after designated cleansing processes.

## Data Availability

Not applicable.

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
