# Peer review of "The Effect of Combined Atmospheric Plasma/UV Treatments on Improving the Durability of Flame Retardants Applied to Cotton"

_molecules, 2022, doi:10.3390/molecules27248737_

Round 1
Reviewer 1 Report
The manuscript shows promising research concerning improving the durability of flame retardants to textiles, which is a very relevant topic.
A lot of data is provided in the manuscript, which in itself is a good thing. However, due to the high amount of different conditions (untreated, plasma-treated, (no) laser, different gasses, different FRs and water-soaking) the text can feel chaotic at times and the way it is written now makes it difficult to keep a sound view of which sample/condition the authors are discussing.
Some methodological or editorial things to change:
Line 138-139: please change the phrasing of this sentence. ‘Decreased to significantly higher levels’ does not sound logical.
Section 2.1.2: a lot of data is provided, but the authors provide little to no explanation about the reason for these results. For example, why is the mass loss for cotton fabric without flame retardant much higher after plasma treatment without laser than with the UV laser? Are these results that can be expected? And why (not)?
Line 270-276: Since there is no clear statistical difference in the phosphorus concentrations between the plasma-treated and non-plasma-treated samples (due to the relatively high error margin) I would refrain from drawing/suggesting conclusions on the effect of the plasma treatment. There is no basis for saying something about the effect of the plasma treatment. Moreover, when looking only to these results and when taking into account the error on XPS-measurements (and EDX-measurements), one has to wonder if there is indeed any phosphorus present on the surface after water-soaking and if the remaining N and Si signals are not just coming from the APTS on the surface.
This is much better explained by the authors in lines 579 – 612. It might be good to already refer to this explanation earlier.
Line 756: the authors want to mention 3 different plasma gasses that are used, but twice the same gas (N2 80%/O2 20%) is mentioned. Looking at the rest of the text, I assume this is a typo and should be N2 80%/CO2 20%.
Line 808: There seems to be an error in which equations are subtracted from each other as equation (2) concerns power dosage.
Reviewer 2 Report
The manuscript from Horrocks and coworkers reports the effect of combined atmospheric plasma/UV laser to improving the durability of flame retardants applied to cotton. In its current form the manuscript is quite good and the results are solidly confirmed no much comments can be done in this regard. After modifying the following minor points, the excellent manuscript should be ready for publication.
1. Page 6, line 208, please explain in detail how EDX results support FR retention percentage.
2. Page 10, line 305, it should be verified whether APTS is successfully introduced and the changes of functional groups should be analyzed in combination with infrared.
3. Page 26, line 859, how to increase the durability of fabrics soaked in water? Please specify how durability is reflected.
4. How about the mechanical property, whiteness, air permeability of the sample?
5. What about this mothed of combined atmospheric plasma/UV laser applied to polyester fabric? Why the flame retardants (TEOS, DAP, APTS, GM, DBAP) are used? Whether other FRs can be used to improving the durability?
6. Which field will the prepared sample be used in? Cloth? Curtain? Because the improving treatment were carried out, and the cost will be high, and the practical application will be seriously limited.
7. Pls improve the Figures quality and clarity, such as Fig. 4 and 7.
Author Response
Please see the appended file

Reviewer 3 Report
(1) Reference mark should make [1,2,3], [2,3,4,5,6,7,8,9,10,11] into [1~3], [2~11]. If this is required by the journal, please ignore my question.
(2) Figure 7 needs to be embellished.
(3) The reference format needs to be unified.
(4) Why don't the materials and methods section come before the results section?
Author Response
Please see the appended file
